# Structural basis for capsid recruitment and coat formation during HSV-1 nuclear egress

Elizabeth B Draganova[1], Jiayan Zhang[2,3,4], Z Hong Zhou[2,3,4], Ekaterina E Heldwein[1]*

[1]Department of Molecular Biology and Microbiology, Tufts University School of Medicine, Boston, United States; [2]Department of Microbiology, Immunology & Molecular Genetics, University of California, Los Angeles (UCLA), Los Angeles, United States; [3]Molecular Biology Institute, UCLA, Los Angeles, United States; [4]California NanoSystems Institute, UCLA, Los Angeles, United States

**Abstract** During herpesvirus infection, egress of nascent viral capsids from the nucleus is mediated by the viral nuclear egress complex (NEC). NEC deforms the inner nuclear membrane (INM) around the capsid by forming a hexagonal array. However, how the NEC coat interacts with the capsid and how curved coats are generated to enable budding is yet unclear. Here, by structure-guided truncations, confocal microscopy, and cryoelectron tomography, we show that binding of the capsid protein UL25 promotes the formation of NEC pentagons rather than hexagons. We hypothesize that during nuclear budding, binding of UL25 situated at the pentagonal capsid vertices to the NEC at the INM promotes formation of NEC pentagons that would anchor the NEC coat to the capsid. Incorporation of NEC pentagons at the points of contact with the vertices would also promote assembly of the curved hexagonal NEC coat around the capsid, leading to productive egress of UL25-decorated capsids.

*For correspondence:
katya.heldwein@tufts.edu

Competing interests: The authors declare that no competing interests exist.

## Introduction

To replicate, all viruses must assemble their progeny virions and release them from the cell while overcoming many obstacles, including cellular compartmentalization. Viruses are thus experts at hijacking, manipulating, and, sometimes, even remodeling cellular architecture during viral morphogenesis and egress. Identifying and understanding the unique aspects of virus-induced cellular remodeling could unveil targets for therapeutic intervention; yet, we are only beginning to understand the mechanisms behind many of these processes.

One prominent example of virus-induced remodeling of cellular architecture can be observed during egress of herpesviruses – enveloped, double-stranded DNA viruses that infect a wide range of hosts, from mollusks to humans. All herpesviruses can establish lifelong, latent infections within the host, from which they can periodically reactivate, spreading to uninfected tissues and hosts and causing a number of ailments. When the virus actively replicates during a primary infection or reactivation of a latent infection, the progeny virions are assembled and released from the cell in a process termed egress whereby herpesvirus capsids traverse several cellular membranes (reviewed in *Bigalke and Heldwein, 2016*; *Johnson and Baines, 2011*; *Roller and Baines, 2017*). First, nuclear capsids bud at the inner nuclear membrane (INM) forming enveloped vesicles that pinch off into the perinuclear space. These perinuclear enveloped virions fuse with the outer nuclear membrane, which releases the capsids into the cytosol. Cytoplasmic capsids then bud again at vesicles derived from the *trans*-Golgi network and early endosomes (reviewed in *Johnson and Baines, 2011*) to form mature, infectious virions that are released from the cell by exocytosis. Whereas many enveloped

viruses acquire their lipid envelopes by budding at the cytoplasmic membranes or the plasma membrane, herpesviruses are unusual among vertebrate viruses in their ability to bud at the nuclear envelope (*Bigalke and Heldwein, 2016*).

Capsid budding at the nuclear envelope requires two conserved herpesviral proteins, which are named UL31 and UL34 in herpes simplex virus type 1 (HSV-1), that form the nuclear egress complex (NEC) (reviewed in *Bigalke and Heldwein, 2016*; *Bigalke and Heldwein, 2017*; *Mettenleiter et al., 2013*). The NEC heterodimer is anchored at the INM through the single C-terminal transmembrane helix of UL34 and faces the nucleoplasm (*Shiba et al., 2000*). UL31 is a nuclear phosphoprotein that colocalizes with UL34 (*Chang and Roizman, 1993*; *Reynolds et al., 2001*) and interacts with the capsid during nuclear egress (*Trus et al., 2007*; *Yang and Baines, 2011*). Both UL31 and UL34 are necessary for efficient nuclear egress, and in the absence of either protein, capsids accumulate in the nucleus and viral replication is reduced by several orders of magnitude (*Fuchs et al., 2002*; *Roller et al., 2000*).

Previously, we discovered that HSV-1 NEC has an intrinsic ability to deform and bud membranes by demonstrating that purified recombinant NEC vesiculates synthetic lipid bilayers in vitro without any additional factors or chemical energy (*Bigalke et al., 2014*). Similar findings were reported with the NEC homolog from a closely related pseudorabies virus (PRV) (*Lorenz et al., 2015*). Using cryogenic electron microscopy and tomography (cryoEM/ET), we showed that the NEC forms hexagonal 'honeycomb' coats on the inner surface of budded vesicles formed in vitro (*Bigalke et al., 2014*). Very similar hexagonal coats were observed in capsidless perinuclear vesicles formed in vivo, in uninfected cells expressing PRV NEC (*Hagen et al., 2015*). Additionally, HSV-1 NEC formed a hexagonal lattice of the same dimensions in crystals (*Bigalke and Heldwein, 2015*). The high-resolution crystal structure of the hexagonal NEC lattice revealed interactions at the lattice interfaces (*Bigalke and Heldwein, 2015*), and subsequent work confirmed that mutations that disrupt oligomeric interfaces reduce budding in vitro (*Bigalke and Heldwein, 2015*; *Bigalke et al., 2014*) and in vivo (*Arii et al., 2019*; *Roller et al., 2010*). Collectively, these findings established the NEC as a viral budding machine that generates negative membrane curvature by oligomerizing into a hexagonal coat on the surface of the membrane.

What remains unclear, however, is how the NEC achieves appropriate coat geometry compatible with negative membrane curvature formation during budding. A purely hexagonal arrangement is flat, so curvature is typically achieved either by insertions of pentagons as found at 12 vertices of an icosahedron (*Zandi et al., 2004*), or by inclusion of irregular defects as observed in several viral coats (*Briggs et al., 2009*; *Heuser, 2005*; *Hyun et al., 2011*; *Schur et al., 2015*). It is tempting to speculate that the capsid geometry may influence the geometry of the NEC coat. In perinuclear visualized in infected cells, the NEC coats appear to be tightly associated with the capsid (*Hagen et al., 2015*). Capsid interactions with the NEC during nuclear budding may be mediated by binding of the capsid protein UL25 to UL31 (*Yang and Baines, 2011*; *Yang et al., 2014*). Moreover, UL25 forms pentagonal complexes at the vertices of the icosahedral herpesvirus capsids (*Dai and Zhou, 2018*; *Furlong, 1978*). However, it is unknown whether interaction with a mature capsid could promote pentagonal formation within NEC coats and if so, how the NEC coat would be arranged around the capsid.

A fortuitous observation that HSV-1 UL25, a capsid protein that decorates the vertices, co-localizes with synthetic liposomes in vitro in the presence of the NEC prompted us to investigate interactions between UL25 and NEC and the effect of UL25 on NEC-mediated budding in vitro. Here, by confocal microscopy, we show that free UL25 (i.e., not on capsid vertices) inhibits NEC-mediated budding in vitro. 3D visualization of the molecular architecture by cryoET further revealed that free UL25 forms a loosely linked net of five-pointed stars on top of membrane-bound NEC layer that may block budding by preventing membrane-bound NEC coats from undergoing conformational changes required for budding. We also found that the NEC forms an alternative pentagonal, rather than hexagonal, arrangement when bound to UL25, and that this phenomenon requires residues 45–73 that form the UL25/UL25 helical bundles on the native capsids. We hypothesize that during nuclear budding, NEC pentagons are formed at the points of contact with the capsid vertices and that they both help anchor the NEC coat to the capsid and generate appropriate coat curvature through the inclusion of pentagons into a hexagonal coat as it assembles around the capsid. This mechanism would ensure successful budding and egress of the UL25-decorated viral capsid.

## Results

### Generation of UL25 variants

HSV-1 UL25 can be expressed in soluble form in *E. coli* only when residues 1–44 are deleted (*Bowman et al., 2006*). Residues 1–50 are necessary and sufficient for capsid binding (*Cockrell et al., 2009*), and in the cryoEM structure of the HSV-1 capsid, these residues mediate extensive interactions with another capsid protein, UL17 (*Dai and Zhou, 2018*). This suggests that these residues are likely disordered in free UL25, potentially leading to aggregation and poor solubility. Therefore, we generated and expressed an HSV-1 UL25Δ44 construct, which lacks residues 1–44. UL25Δ44 was soluble and could be purified, in agreement with the previous report (*Bowman et al., 2006*), but was proteolytically cleaved during purification despite the presence of protease inhibitors (*Figure 1b*). N-terminal sequencing of the single proteolytic product by Edman degradation yielded the amino acid sequence AAELPV, corresponding to residues 73–78. This suggested that UL25Δ44 was cleaved between residues Q72 and A73. To prevent heterogeneity due to cleavage, we generated two constructs: UL25Δ44 Q72A, which has a single point mutation designed to eliminate the cleavage site, and UL25Δ73, which corresponds to the cleavage product. Both constructs yielded a single UL25 species after purification (*Figure 1b*).

### UL25Δ44 Q72A inhibits NEC-mediated budding

To assess the effect of UL25 on NEC-mediated budding, we used an established in-vitro budding assay utilizing recombinant, soluble NEC220 (full-length UL31 and UL34 residues 1–220), fluorescently labelled giant unilamellar vesicles (GUVs), and membrane-impermeable fluorescent dye, Cascade Blue (*Bigalke et al., 2014*). NEC220 and UL25Δ44 Q72A were added to the GUVs in 1:1, 1:6, 1:8, 1:10, or 1:20 molar ratios, and budding events were quantified. UL25Δ44 Q72A inhibited NEC-mediated budding in a dose-dependent manner, and at 1:10 or 1:20 NEC:UL25 molar ratios, few budding events were observed (*Figure 1c*). By contrast, UL25Δ73 did not inhibit budding even at a 1:20 ratio of NEC:UL25 (*Figure 1c*), which suggested that residues 45–73 were necessary for inhibition.

UL25Δ44 consists of a long N-terminal α-helix (residues 48–94), followed by a flexible linker unresolved in the cryoEM structure, and a C-terminal globular core (residues 134–580) (*Figure 1a*). Residues 45–73 encompass the N-terminal half of the long α-helix. To further narrow down the inhibitory region within UL25, we analyzed its sequence conservation. Sequence alignment of UL25 homologs from five alphaherpesviruses revealed a divergent N terminus followed by a highly conserved alanine-rich region, residues 61–69 (*Figure 1a*). We generated the UL25Δ58 Q72A construct lacking the divergent N terminus of the α-helix (*Figure 1b*). UL25Δ58 Q72A did not inhibit NEC220 budding (*Figure 1c*). We also generated a UL25Δ50 Q72A construct (as a control for studies using eGFP-UL25 chimera described below), which inhibited budding at a 1:10 NEC:UL25 ratio, the minimal UL25 concentration for budding inhibition (*Figure 2a*). Thus, residues 51–73 appear essential for inhibition, whereas residues 45–50 are dispensable.

### UL25 does not bind synthetic membranes

We first tested whether UL25 inhibited NEC-mediated budding by competing with the NEC for binding to membranes. We utilized an established co-sedimentation assay utilizing multilamellar vesicles (MLVs) of the same composition as the GUVs used in the budding assay (*Bigalke et al., 2014*). Unlike NEC220, UL25Δ44 Q72A did not bind synthetic lipid vesicles (*Figure 1d*) and, therefore, could not compete with the NEC220 for binding to membranes.

### UL25Δ44 and NEC do not interact in solution

UL25 does not bind membranes (*Figure 1d*), so, to inhibit NEC-mediated budding, UL25 must instead bind to the NEC. However, no binding was detected in solution, either between UL25Δ44 and NEC220 by isothermal titration calorimetry (*Figure 1—figure supplement 1*) or between UL25Δ44 and NEC185Δ50 [a truncated construct that was crystallized previously (*Bigalke and Heldwein, 2015*)] by size-exclusion chromatography (*Figure 1—figure supplement 1*). Therefore, to bind UL25, NEC may need to be bound to the membrane. Surface plasmon resonance experiments

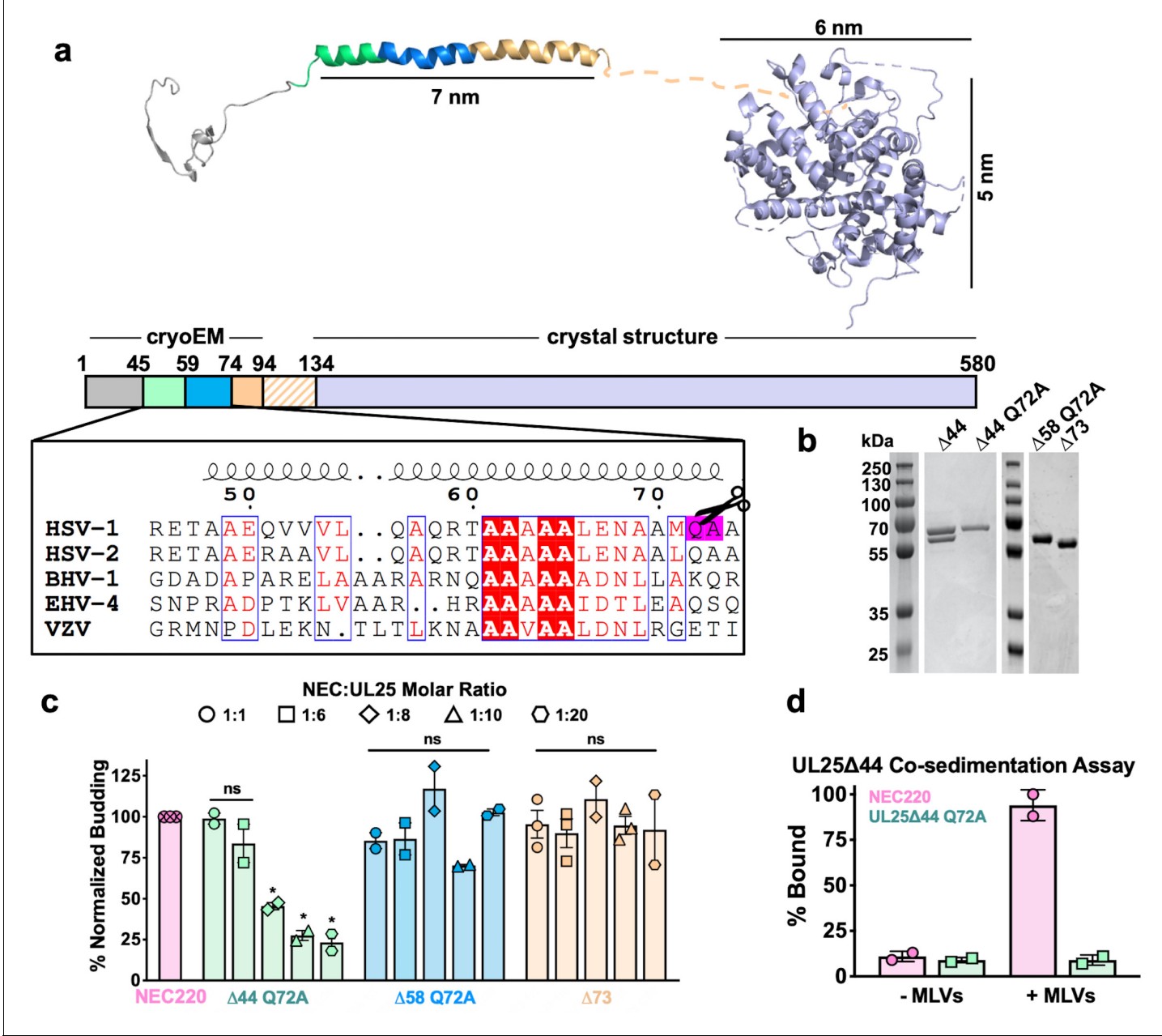

**Figure 1.** Inhibition of NEC-mediated budding by UL25 constructs. (a) The UL25 structure and a diagram of domain organization is shown along with a multiple sequence alignment of UL25 residues 45–74 from five alphaherpesviruses. Sequence alignment was generated using Clustal Omega[45] and displayed using ESPript 3.0[46]. Identical residues are shown as white letters on a red background. Similar residues are shown as red letters in a blue box. Secondary structure derived from the cryoEM reconstruction of capsid-bound HSV-1 UL25 is shown above the alignment. The following herpesvirus sequences were used (GenBank GeneID numbers in parentheses): HSV-1, herpes simplex virus type 1, strain 17 (2703377); HSV-2, herpes simplex virus type 2, strain HG52 (1487309); BHV-1, bovine herpesvirus-1 (4783418); EHV-4, equine herpesvirus-4, strain NS80567 (1487602); and VZV, varicella-zoster virus, strain Dumas (1487687). (b) SDS-PAGE of purified UL25 constructs: UL25Δ44 (cleaved product; 57 kDa), UL25Δ44 Q72A (single product; 57 kDa), UL25Δ58 Q72A (56 kDa) and UL25Δ73 (54 kDa). (c) UL25Δ44 Q72A inhibits NEC budding, whereas other UL25 constructs do not. For each condition, NEC-mediated budding was tested at 1:1, 1:6, 1:8, 1:10, and 1:20 NEC:UL25 molar ratios. Each construct was tested in at least two biological replicates, consisting of three technical replicates. Symbols show average budding efficiency of each biological replicate relative to NEC220 (100%; pink). Error bars represent the standard error of measurement for at least two individual experiments. Significance compared to NEC220 was calculated using an unpaired t-test against NEC220. *p-value<0.1. The source file with all raw data values is provided in *Figure 1—source data 1*. (d) UL25Δ44 Q72A does not bind to acidic lipid membranes.

The online version of this article includes the following source data and figure supplement(s) for figure 1:

*Figure 1 continued on next page*

*Figure 1 continued*

**Source data 1.** Raw data and background values collected for GUV budding assays of NEC220 in the presence of either UL25Δ44 Q72A, UL25Δ58 Q72A or UL25Δ73.
**Figure supplement 1.** NEC-UL25 binding studies.

were also performed, but significant nonspecific binding precluded clear data interpretation (*Figure 1—figure supplement 1*).

## Both inhibitory UL25Δ44 Q72A and non-inhibitory UL25Δ73 colocalize with membranes in the presence of the NEC

To visualize UL25 localization in the presence of NEC and membranes by confocal microscopy, we generated the eGFP-tagged versions of the inhibitory and non-inhibitory UL25 constructs, eGFP-UL25Δ44 Q72A and eGFP-UL25Δ73. However, eGFP-UL25Δ44 Q72A construct was unstable during purification, so eGFP-UL25Δ50 Q72A was generated instead. eGFP-UL25Δ50 Q72A (as well as its untagged version UL25Δ50 Q72A) efficiently inhibited NEC-mediated budding, whereas eGFP-UL25Δ73 did not (*Figure 2a*). When eGFP-tagged UL25 constructs were incubated with the fluorescently labelled GUVs, no eGFP signal was detected on the GUV membranes (*Figure 2—figure supplement 1*), confirming that UL25 did not bind membranes directly.

Next, eGFP-UL25Δ50 was incubated with the GUVs in the presence of the NEC220, at a 1:10 molar ratio of NEC to UL25 (the minimal inhibitory UL25 concentration). In the presence of the NEC220, the eGFP-UL25Δ50 colocalized with the GUV membranes, and very little budding was detected (*Figure 2b*). UL25 itself does not bind membranes, so, instead, it must be binding NEC that is bound to the surface of the GUVs.

In the presence of the NEC220, eGFP-UL25Δ73 also colocalized with the GUV membranes. In this case, the eGFP signal was sometimes detected on the membranes of intraluminal vesicles (ILVs) inside the GUVs (*Figure 2c*) – a product of budding – which confirmed that binding of eGFP-UL25Δ73 to the NEC220 did not interfere with budding (*Figure 2a*) and that eGFP-UL25Δ73 could even remain bound to the NEC-coated membranes throughout budding.

In many cases, however, the eGFP-UL25Δ73 was clustered around the unbudded GUVs, probably due to its aggregation (*Figure 2d*). Such aggregation was not observed for eGFP-UL25Δ50 Q72A (*Figure 2b*). It is conceivable that eGFP-UL25Δ73 aggregates on NEC-coated GUVs because it lacks half of the long N-terminal helix of UL25 (*Figure 1a*). Although such aggregation inhibits budding locally (*Figure 2d*), bulk measurements show that NEC-mediated budding remains efficient in the presence of eGFP-UL25Δ73 (*Figure 2a*). We hypothesize that sequestration of large amounts of aggregated eGFP-UL25Δ73 on a few NEC-coated GUVs reduces its concentration throughout the sample, allowing budding to proceed. Taken together, these results suggested that while both inhibitory and non-inhibitory UL25 constructs could bind the membrane-bound NEC, the binding of the inhibitory UL25 construct blocked NEC-mediated budding, whereas the binding of the non-inhibitory UL25 construct did not.

## Mutations within the putative capsid-binding site on the NEC reduce the extent of UL25 inhibition

Residues D275, K279, and D282 at the membrane-distal tip of UL31 have been implicated in capsid binding in PRV (*Rönfeldt et al., 2017*) and in HSV-1 (*Takeshima et al., 2019*). We generated a quadruple UL31 mutant in which D275, K279, D282, and a nearby C278 were replaced with alanines. The corresponding mutant NEC220, termed capsid-binding mutant (NEC220-CBM), mediated budding at levels similar to the WT NEC220 (*Figure 3a*) but was much less sensitive to inhibition by UL25Δ44 Q72A (*Figure 3a*) as demonstrated by the 25% reduction in budding of NEC220-CBM compared to 75% for NEC220 (*Figure 1c*). Moreover, eGFP-UL25Δ50 Q72A did not co-localize with the GUV membranes in the presence of the NEC220-CBM (*Figure 3b*). These results suggested that UL25Δ44 Q72A bound to the membrane-distal tip of UL31 and that this interaction was essential for its inhibitory activity.

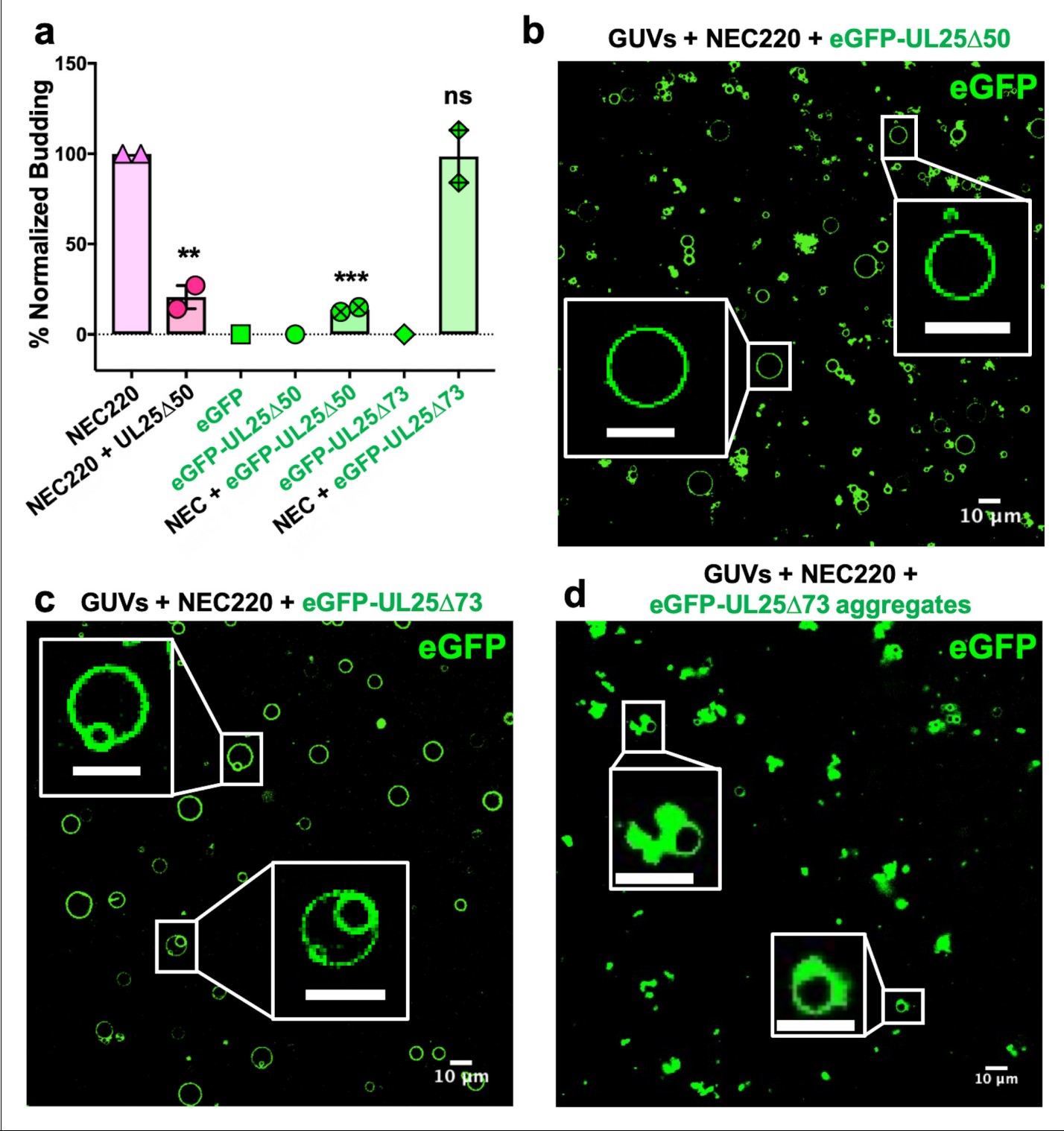

**Figure 2.** eGFP-UL25Δ50 inhibits NEC budding while eGFP-UL25Δ73 does not. (a) Quantification of NEC budding in the presence of either eGFP-UL25Δ50 or eGFP-UL25Δ73. Each construct (except in the absence of NEC220) was tested in at least two biological replicates, each consisting of three technical replicates. Symbols show the average budding efficiency of each biological replicate relative to NEC220 (100%). Error bars represent the standard error of measurement for at least two individual experiments. Significance compared to NEC220 was calculated using an unpaired t-test against NEC220. **p-value<0.01 and ***p-value<0.001. The source file with all raw data values is provided in *Figure 2—source data 1*. (b) Confocal image of eGFP-UL25Δ50 bound to NEC-coated vesicles. No budding is observed. (c) Confocal image of eGFP-UL25Δ73 either bound to or budded into vesicles with the NEC. (d) Confocal image of eGFP-UL25Δ73 aggregating on the surface of NEC-coated vesicles. All scale bars = 10 μm.

*Figure 2 continued on next page*

*Figure 2 continued*

The online version of this article includes the following source data and figure supplement(s) for figure 2:

**Source data 1.** Raw data and background values collected for GUV budding assays of NEC220 in the presence of either UL25Δ50 Q72A, eGFP-UL25Δ50 Q72A or eGFP-UL25Δ73.

**Figure supplement 1.** Confocal image of GUVs (red) and eGFP-UL25Δ50 Q72A (green) showing that eGFP-UL25Δ50 Q72A does not bind GUV membranes.

## UL25 binds membrane-bound NEC

To understand how UL25 inhibits NEC-mediated budding, we turned to cryoEM. Previously, we showed that NEC-mediated budding of synthetic large unilamellar vesicles (LUVs) resulted in the formation of smaller vesicles containing ~11 nm thick internal NEC coats (*Bigalke et al., 2014*). Here, UL25Δ44 Q72A and NEC220 (at a 1:10 molar ratio of NEC to UL25) were incubated with LUVs of the same composition as the GUVs used in the budding assay and visualized by cryoEM. In the presence of UL25Δ44 Q72A and NEC220, the LUVs were mostly spherical, and their external surface was coated with ~17 nm thick coats (*Figure 4a*), although these typically did not cover the entire surface

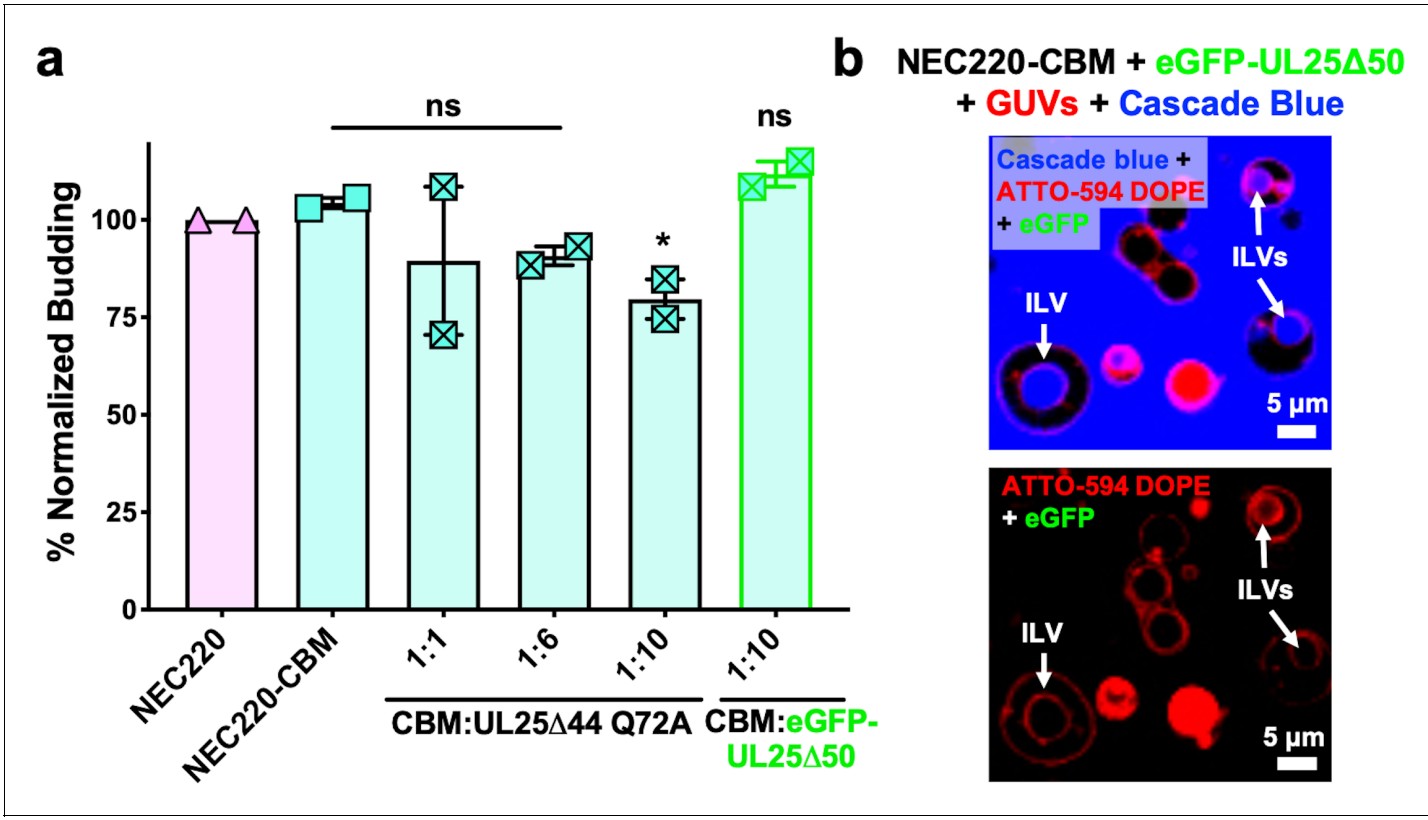

**Figure 3.** UL25 inhibits NEC220-CBM budding to a lesser extent. (a) NEC220-CBM budding is not inhibited to the same extent as NEC220 budding by either UL25Δ44 Q72A or eGFP-UL25Δ50 Q72A. Budding was tested at 1:1, 1:6 and 1:10 NEC220-CBM:UL25 molar ratios for UL25Δ44 Q72A and at a 1:10 NEC-CBM:UL25 molar ratio for eGFP-UL25Δ50 Q72A. Each condition was tested in at least two biological replicates, each consisting of three technical replicates. Symbols represent average budding efficiency of each biological replicate relative to NEC220 (100%). Error bars represent the standard error of measurement for at least two individual experiments. Significance compared to NEC220 was calculated using an unpaired t-test against NEC220. *p-value<0.1. The source file with all raw data values is provided in *Figure 3—source data 1*. (b) Confocal microscopy image showing eGFP-UL25Δ50 Q72A does not bind to NEC220-CBM coated GUVs as indicated by the lack of green signal on the membranes of intraluminal vesicles (ILVs) formed by NEC220-CBM budding (indicated by white arrows). Top panel shows red (ATTO-594 DOPE), green (eGFP), and blue (Cascade Blue) channels. Bottom panel shows red (ATTO-594 DOPE) and green (eGFP) channels only. Scale bars = 5 μm.

The online version of this article includes the following source data for figure 3:

**Source data 1.** Raw data and background values collected for GUV budding assays of NEC220-CBM in the presence of either UL25Δ44 Q72A or GFP-UL25Δ50 Q72A.

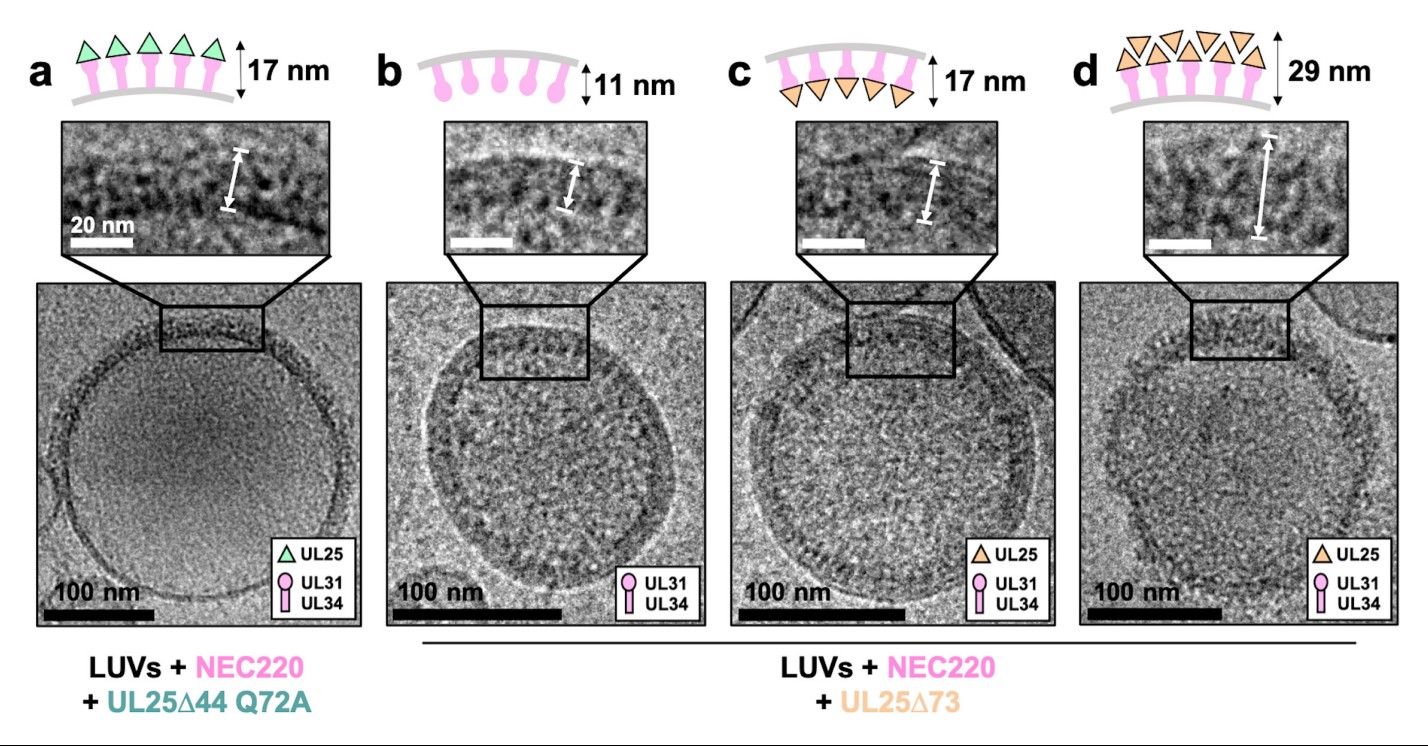

**Figure 4.** CryoEM shows UL25Δ44 Q72A inhibits NEC220 budding while UL25Δ73 does not. (a) UL25Δ44 Q72A bound to the NEC220 on the outside of the unbudded lipid vesicles, forming a fence-like array (~17 nm). In the presence of UL25Δ73, three scenarios have been observed: (b) NEC220 alone bound to the inner surface of the budded lipid vesicles (~11 nm); (c) UL25Δ73 bound to the NEC220, which is itself bound to the inner surface of the budded lipid vesicles (~17 nm), and (d) UL25Δ73 aggregates bound to the NEC on the outside of the unbudded lipid vesicles (>29 nm). Budded lipid vesicles in panels b and c are no longer contained within a 'mother' lipid vesicle and represent the end-product of budding. Scale bars = 100 nm. Inset scale bars = 20 nm. All inset panels are shown on the same scale. White arrows in insets define measurement boundaries of vesicle-bound proteins displayed in the corresponding cartoon models.

The online version of this article includes the following figure supplement(s) for figure 4:

**Figure supplement 1.** Incomplete distribution of NEC-UL25 around vesicles.

**Figure supplement 2.** Slices of selected tomograms of NEC220/UL25Δ73-bound vesicles.

(*Figure 4a* and *Figure 4—figure supplement 1*). The external coats formed in the presence of UL25Δ44 Q72A were ~6 nm thicker than the internal NEC coats, and the diameter of the globular portion of UL25 is also ~6 nm. Therefore, we hypothesize that the external coats are composed of a UL25Δ44 Q72A layer positioned on top of the membrane-bound NEC220 layer (*Figure 4a*). Very few budded vesicles were observed under these conditions, which is consistent with the inefficient budding observed by confocal microscopy (*Figure 1c*). Thus, binding of UL25Δ44 Q72A to the NEC220 on the surface of the lipid vesicles correlated with its ability to inhibit NEC-mediated budding.

Co-incubation of UL25Δ73 and NEC220 with LUVs yielded budded vesicles (*Figure 4b and c*), some of which contained ~17 nm thick internal coats (*Figure 4c*), presumably containing UL25Δ73 bound to the NEC220. We also observed budded vesicles containing ~11 nm thick internal coats (*Figure 4b*), presumably containing only NEC220, in accordance with our previous report (*Bigalke et al., 2014*). Last but not least, we observed unbudded LUVs containing >29 nm thick heterogeneous protein aggregates on the external surface (*Figure 4d* and *Figure 4—figure supplement 2*) that were similar to UL25Δ73 aggregates observed by confocal microscopy (*Figure 2d*) in the sense that both ranged in thickness and distribution of the protein aggregates around the vesicles. This aggregation precluded cryoET data collection on LUVs coated with UL25Δ73 and NEC220 as described below for UL25Δ44 Q72A and NEC220.

## UL25Δ44 Q72A forms a net of stars bound to NEC pentagons

Interactions between UL25Δ44 Q72A and membrane-bound NEC220 were visualized in three dimensions by cryoET (*Figure 5*). Sub-tomographic averaging of the 3D reconstructions of unbudded LUVs coated with NEC220 and UL25Δ44 Q72A (*Figure 5a*) revealed that UL25Δ44 Q72A formed a net of five-pointed stars (*Figure 5c*) covering the surface of membrane-bound NEC220 that, in turn, formed pentagons (*Figure 5d*). Five-pointed stars of UL25 were positioned directly on top of the NEC pentagons (*Figure 5c,d*). The pentagonal arrangement was observed during averaging even prior to imposing five-fold symmetry (*Figure 5—figure supplement 1*). Within the NEC220/ UL25Δ44 Q72A dataset, we only observed five-fold symmetry in the protruding densities, whereas in the NEC220 only dataset, we observed only six-fold symmetry (*Figure 5b*). Therefore, as far as we can tell, when bound to UL25Δ44 Q72A, the NEC220 only forms pentagons and is not a mixture of pentagons and hexagons.

The ability of the NEC to form pentagons was unexpected because prior to that point, only hexagonal NEC coats had been observed on budded vesicles formed in vitro (*Figure 5b*; *Bigalke et al., 2014*) and on perinuclear vesicles formed in vivo in NEC-expressing uninfected cells (*Hagen et al., 2015*). The NEC pentagons and hexagons have similar dimensions, ~10.5 nm vs. ~11 nm in width (*Figure 5b,d*) with ~6.5 nm vs. ~6.3 nm sides (*Figure 5d*; *Bigalke et al., 2014*). From the crystal structure, we know that the hexagons are hexamers of the NEC heterodimers (*Bigalke and Heldwein, 2015*). Therefore, we hypothesize that the pentagons are pentamers of the NEC heterodimers.

It should be noted that an entirely pentagonal lattice would yield a small spherical object with high curvature and icosahedral symmetry – neither of which were observed in our cryoET averages. Visual analysis of the cryoEM images suggested that UL25Δ44 Q72A/NEC220 spikes did not fully cover the surface of the LUVs, and instead formed patches rather than full coats (*Figure 4a* and *Figure 4—figure supplement 1*). Likewise, mapping the coordinates of each sub-tomogram back onto the raw data also showed that particles selected for analysis only partially cover the vesicles (*Supplementary file 1*). Furthermore, the surrounding pentagons in the sub-tomographic averages (*Figure 5*) were not as sharply defined as the central one, which contributed the most power to alignment during data processing, suggesting a flexible arrangement, possibly due to weak interactions. We hypothesize that the adjacent NEC pentagons capped with UL25 stars are loosely connected, which results in a patchy network of pentagons instead of small, well-defined coats that would form if the arrangement were rigid. Regardless, the cryoET data clearly show the ability of the NEC to form an alternative, pentagonal arrangement in the presence of UL25 thus documenting the ability of the NEC to form different types of oligomers.

## Discussion

The intrinsic ability of the NEC to deform and bud membranes and to oligomerize into a hexagonal coat is well established (reviewed in *Bigalke and Heldwein, 2016*; *Bigalke and Heldwein, 2017*; *Mettenleiter, 2016*; *Roller and Baines, 2017*). However, it is unclear how the capsid triggers the formation of the NEC coat around it or how the NEC coat is anchored to the capsid. Moreover, a purely hexagonal lattice is flat, so it remains unknown how the curvature is generated within the hexagonal NEC coat. Here, we have made two key observations. First, UL25 binds the NEC on lipid vesicles in vitro and inhibits NEC-mediated budding in a dose-dependent manner. Second, the NEC forms pentagons when bound to UL25. We hypothesize that UL25/NEC interactions observed in vitro mimic UL25/NEC interactions that anchor the NEC coat to the capsid vertices. We further hypothesize that NEC pentagons formed at the points of contact with the capsid vertices could nucleate the assembly of the otherwise hexagonal NEC coats and introduce curvature into them.

### UL25 inhibits NEC-mediated budding in vitro by forming a star-shaped net over the membrane-bound NEC layer

We found that UL25Δ44 Q72A construct inhibited NEC220-mediated budding in vitro in a dose-dependent manner. We also observed that this UL25 construct formed five-pointed stars linked into a loose net on the surface of the membrane-bound NEC220 layer. To understand the nature of inhibition by UL25, we modelled UL25 interactions within the stars by examining UL25 interactions on

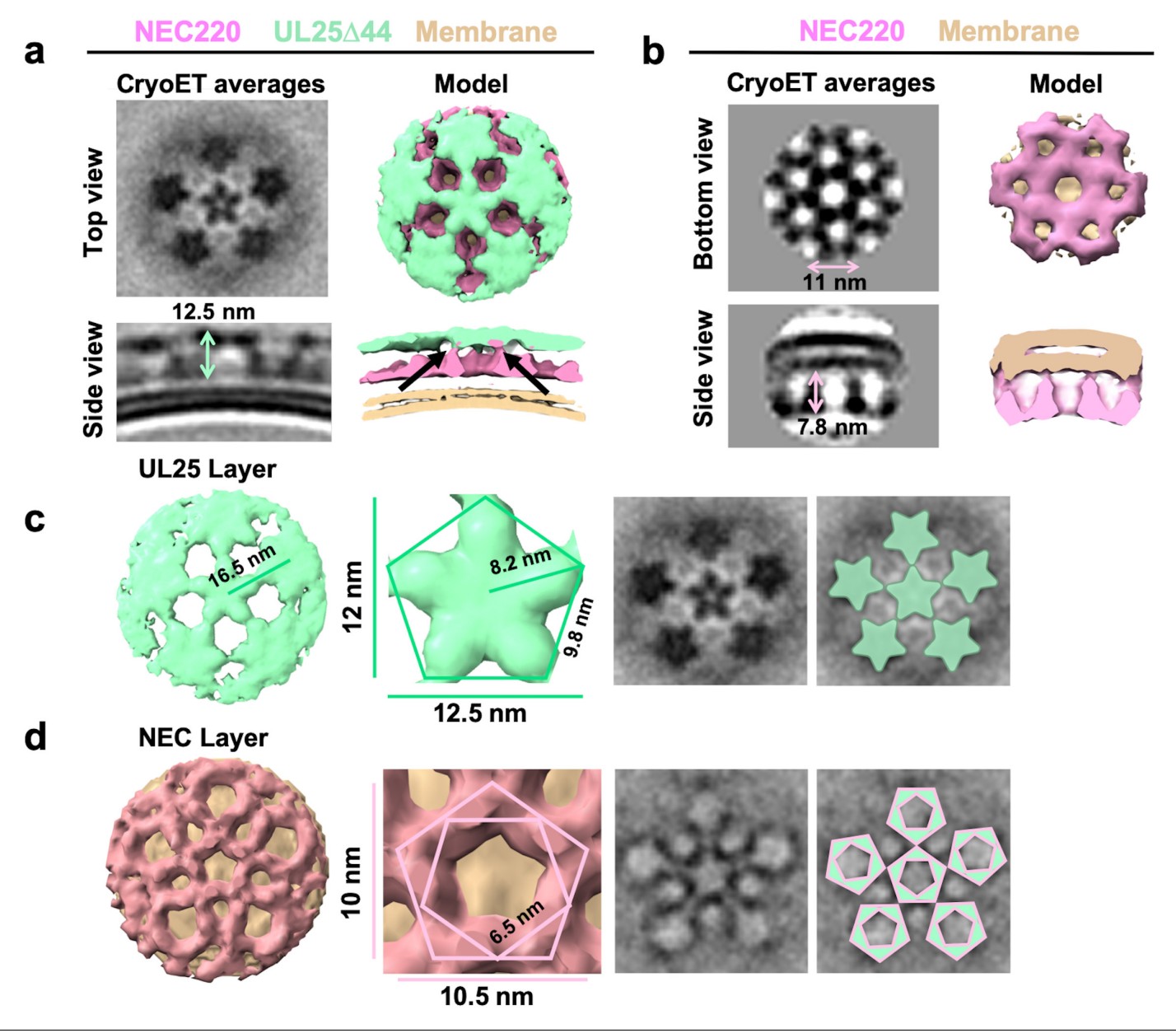

**Figure 5.** CryoET of UL25-mediated inhibition of NEC budding. (a) CryoET averages of NEC220 in the presence of UL25Δ44 Q72A (top and side views). Corresponding 3D models are shown with NEC220 (pink) and UL25Δ44 Q72A (green). The vesicle bilayer is shown in beige. The models show the UL25 layer coating the NEC layer in five-pointed stars on the outside of the vesicles. The length of the NEC-UL25 spikes is 12.5 nm. Black arrows indicate the point of tilt within the NEC layer. (b) CryoET averages of NEC220 forming hexameric lattices in the presence of membranes (bottom and side views). Corresponding 3D models are shown with NEC (pink) and the vesicle bilayer (beige). The diameter of the hexameric rings is ~11 nm, while the length of the spikes is 7.8 nm. (c) CryoET model and averages of the UL25 layer (green) highlighting the five-pointed star formation of UL25 (represented here as a pentamer of dimers) in the presence of NEC. (d) CryoET model and averages of the NEC layer showing NEC220 forming a pentagonal lattice (pink pentagons), rather than hexagonal (as seen for wild-type in panel b). Green triangles indicate location of UL25 binding to the NEC.

The online version of this article includes the following figure supplement(s) for figure 5:

**Figure supplement 1.** CryoET averages of NEC220 in the presence of UL25Δ44 Q72A (top and side views) prior to applying five-fold symmetry.

the capsids. The five-pointed UL25 stars formed in vitro resemble the five-pointed stars crowning each capsid vertex that are composed of five copies of the capsid-associated tegument complex (CATC) (*Dai and Zhou, 2018*; *Figure 6a*). Each CATC, in turn, is composed of two copies of UL25, one copy of UL17, and two copies of the C-terminal portion of the tegument protein UL36 (*Dai and*

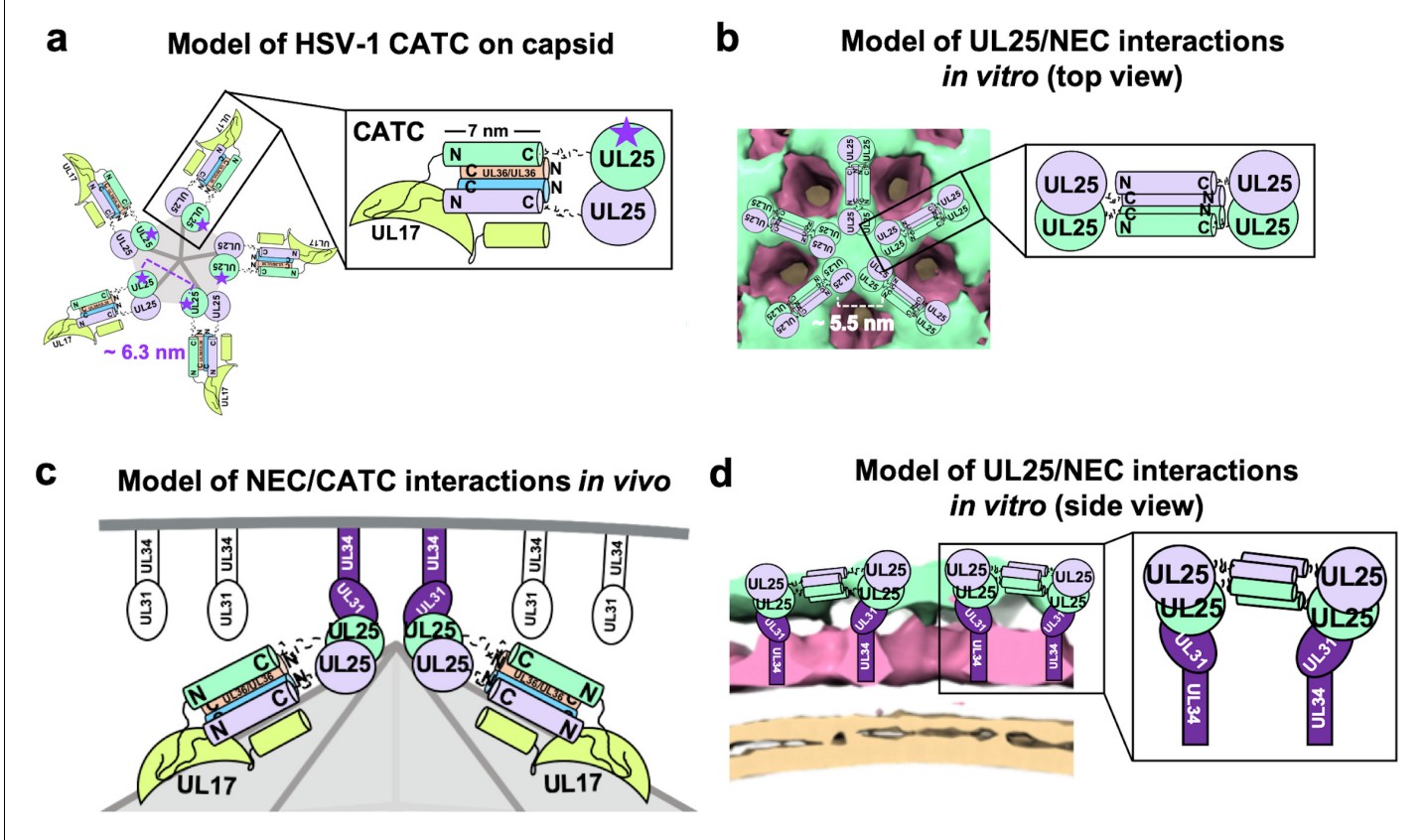

**Figure 6.** Models of UL25/UL25 and UL25/NEC interactions in vitro and in vivo. (**a**) A schematic representation of the pentagonal HSV-1 CATC [two copies of UL25 (green and purple), two copies of C-terminal UL36 (peach and blue) and one copy of UL17 (lime green)] arrangement at the capsid vertex. Inset shows a close-up view of the characteristic antiparallel four-helix bundle composed of two UL25 helices and two UL36 helices. Purple stars indicate the proposed UL25 copies that bind to the NEC upon capsid docking. The distance between the centers of two adjacent inner UL25 cores (green) in the capsid (*Dai and Zhou, 2018*) is ~6.3 nm. (**b**) Proposed model of the UL25 stars formed in vitro. The distance between the centers of two adjacent UL25 dimers is ~5.5 nm. Inset shows a close-up view of the proposed antiparallel four-helix bundle composed of two pairs of UL25 helices from adjacent stars. We hypothesize that four-helix bundles link the neighboring UL25 stars into a net. (**c**) Proposed side-view model of the NEC (purple) interacting with the most surface exposed capsid-bound UL25 (green), resulting in a pentameric NEC (indicated by dark purple coloring). NEC molecules prior to capsid binding are shown in an unknown oligomeric state (white). (**d**) Side view of the proposed NEC/UL25 interactions in vitro.

*Zhou, 2018*) and has a characteristic antiparallel four-helix bundle composed of two UL25 helices and two UL36 helices (*Figure 6a*). Therefore, we hypothesize that UL25Δ44 Q72A also forms an anti-parallel four-helix bundle when bound to the NEC220 on the membrane surface in vitro. However, in this case, the helical bundle is composed of two pairs of UL25 helices from adjacent stars (*Figure 6b*), linking them into a net. Each UL25 'star' would then consist of 10 copies of UL25, with cores arranged in the center and five pairs of helices radiating out (*Figure 6b*).

This model shown in *Figure 6b* implies that UL25 is an oligomer, likely a decamer. We have not yet observed any UL25 oligomerization in solution at the concentrations used in our experiments. While we cannot exclude the possibility that the UL25 can oligomerize in solution at protein concentrations higher than what was used in our experiments, we favor an alternative hypothesis that UL25 oligomerization requires a binding partner – a capsid or an NEC oligomer. The latter notion is supported by our observations that the NEC and UL25 do not interact in solution but do interact when the NEC is tethered to a scaffold, for example, the membrane of a lipid vesicle or the surface of an SPR chip.

We propose that UL25Δ44 Q72A inhibits NEC220-mediated budding in vitro by binding the membrane-bound NEC and forming five-pointed stars on top of NEC pentagons that are linked into a loose net (*Figure 7b*). Such a net could inhibit budding by restricting conformational changes within the NEC lattice necessary to generate negative membrane curvature (*Figure 7a*). By contrast,

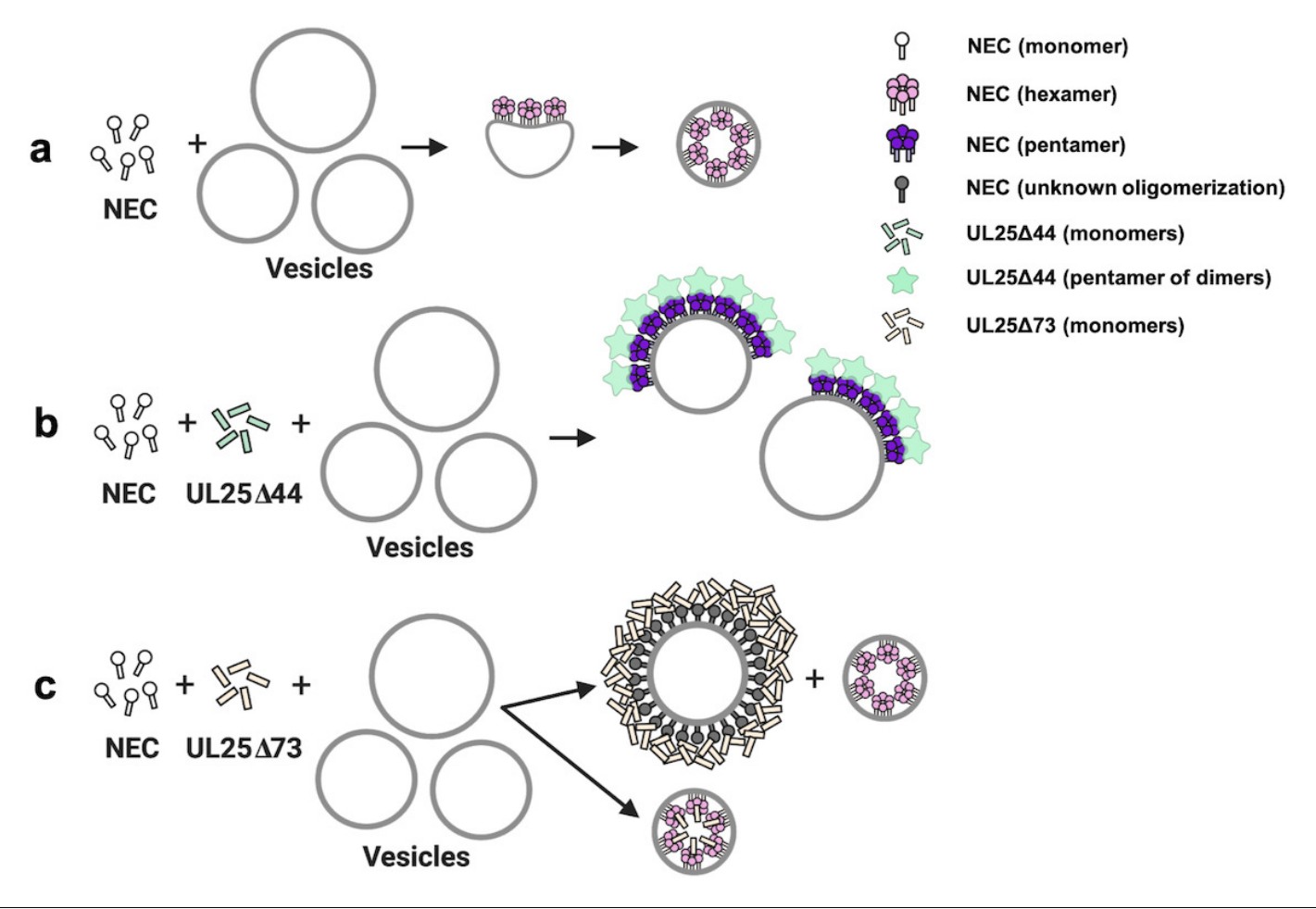

**Figure 7.** A model of NEC-mediated budding in the absence and presence of UL25, in vitro. (**a**) NEC-mediated budding requires only the NEC, which vesiculates membranes by forming hexagonal coats (pink) that, potentially, contain irregular defects to achieve curvature. (**b**) UL25Δ44 Q72A (green) inhibits NEC-mediated budding by inducing the formation of a pentagonal NEC coat (purple) suboptimal for budding. (**c**) UL25Δ73 (peach) aggregates around some NEC-coated vesicles, which blocks budding. Sequestration of UL25Δ73 at a few locations reduces its concentration elsewhere and enables budding. Binding of UL25Δ73 to NEC in the absence of aggregation does not interfere with budding, and bound UL25Δ73 buds into vesicles with the NEC. This figure was created with Biorender.com.

UL25Δ73 construct does not inhibit budding because it has a much shorter N-terminal helix (*Figure 1a*) and cannot form a four-helix bundle. UL25Δ73 is also prone to aggregation likely because the shorter helix is less stable (*Figure 7c*).

It is tempting to speculate that free UL25, which is likely present within the nucleus, could inhibit the budding activity of the NEC during infection. Both in vitro and in NEC-expressing uninfected cells, the NEC is constitutively active. However, during infection, the NEC budding activity is controlled, presumably by a viral protein, to ensure budding of only mature capsids and to prevent premature, non-productive budding. It would be interesting to test whether co-expression of UL25 along with the NEC could inhibit NEC-mediated budding.

## UL25/NEC interactions in vitro may mimic interactions between CATC at the capsid vertices and the NEC coats during infection

How the NEC coat is anchored to the capsid is yet unclear. Nevertheless, interactions between UL25 and UL31 have been implicated in this process. UL31 from HSV-1 infected cell lysates was shown to co-immunoprecipitate with UL25 (*Yang and Baines, 2011*) and bind nucleocapsids (*Yang et al., 2014*). More recently, NEC185Δ50 (a previously crystallized truncated construct with an intact

membrane-distal UL31 region) from *E. coli* was shown to bind purified nucleocapsids from HSV-1 infected cells in a pull-down assay (*Takeshima et al., 2019*). This interaction requires UL25 because the VP5, VP23, and UL17 capsid proteins were only detected if UL25 was present on capsids, suggesting NEC binds capsid-bound UL25. UL25 likely binds to the membrane-distal tip of UL31 because capsid binding to the NEC requires UL31 residues R281 and D282, which are located in this region (*Takeshima et al., 2019*). Other charged residues within the membrane-distal region of UL31 have also been implicated in capsid interactions in PRV (*Rönfeldt et al., 2017*). Our work showed that four mutations within this membrane-distal region (NEC22-CBM) rendered this mutant less sensitive to inhibition by UL25 relative to the WT NEC220. Collectively, these observations implicate interactions between UL25 and the membrane-distal end of UL31 in NEC/capsid interactions during nuclear egress.

In our cryoET reconstruction, UL25 cores sit on the membrane-distal ends of the NEC (*Figure 6b*). Therefore, we hypothesize that the interactions between UL25 and membrane-bound NEC that we observed by cryoET in vitro (*Figure 6b and d*) mimic how the NEC interacts with UL25 on the capsid vertices (*Figure 6a and c*). Although the distance between the presumed locations of the cores within the UL25 stars observed in vitro (*Figure 6b*) is shorter than between the cores of the innermost UL25 copies within the CATC (*Figure 6a*), the cores could move into most favorable orientations for NEC binding. Indeed, in cryoEM reconstruction of the HSV-1 capsid, the cores are connected to the N-terminal helices by long, flexible linkers and appear dynamic (*Dai and Zhou, 2018*). On the other hand, the NEC may be able to tilt relative to the membrane surface (*Figure 5a*), which would also allow each NEC to adopt the optimal orientation for binding the UL25 cores at the capsid vertices. The model shown in *Figure 6a and c* illustrates how the NEC coat may be anchored to the capsid.

## The NEC pentagons could introduce curvature into the hexagonal NEC lattice

The ability of the NEC to oligomerize into a hexagonal lattice in vitro and in vivo is well documented (*Bigalke and Heldwein, 2015*; *Bigalke et al., 2014*; *Hagen et al., 2015*) and is an important feature of its membrane deformation mechanism (*Bigalke and Heldwein, 2015*; *Bigalke et al., 2014*; *Roller et al., 2010*). But how the hexagonal NEC lattice accommodates curvature is yet unclear.

A strictly hexagonal lattice is flat, so the curvature is typically achieved through the inclusion of lattice defects, also termed insertions. These can either be regular insertions of a different geometry, for example, pentagons – as observed in icosahedral or fullerene-like capsids – or irregular insertions. Although no deviations from the hexagonal symmetry have yet been visualized in any NEC coats (*Bigalke et al., 2014*; *Hagen et al., 2015*; *Newcomb et al., 2017*), this could be due to the low resolution of the cryoET reconstructions or the imposition of symmetry in averaging. For example, one study used cryoEM and cryoET to visualize NEC/capsid interactions within perinuclear enveloped virions isolated from cells infected with an US3 kinase-null HSV-1 (*Newcomb et al., 2017*), a mutation that causes the accumulation of perinuclear enveloped virions. Although only the hexagonal NEC arrays were observed, the averaging of NEC/CATC interactions was hindered by significant noise in the relevant regions of the tomograms because the NEC coat did not have the same icosahedral symmetry as the capsid. As the result, NEC/CATC interactions were difficult to visualize. Additionally, the lack of US3 could have altered the structure of the NEC coat or the NEC/CATC interactions.

In perinuclear enveloped virions formed in PRV-infected cells, the NEC coats appear tightly associated with the capsid (*Hagen et al., 2015*). Capsidless perinuclear vesicles formed in uninfected cells expressing PRV NEC (*Hagen et al., 2015*) are relatively uniform in size (~115 nm in diameter; *Figure 8*, inset) but smaller than the capsid (~125 nm in diameter *Dai and Zhou, 2018*; *Liu et al., 2017*) or the perinuclear enveloped virions isolated from cells infected with the HSV-1 US3-null mutant virus (~160 nm in diameter *Newcomb et al., 2017*; *Figure 8*). The capsid thus appears to define the size of NEC-budded vesicles during infection, so the capsid geometry could influence the geometry of the NEC coat.

Here, for the first time, we observed that the NEC can form not only hexagons but also pentagons, the latter when bound to UL25 in vitro. This finding suggests that during nuclear egress, NEC pentagons could form at the points of contact with UL25 (within CATC) at the capsid vertices. The inclusion of pentagons into a hexagonal coat would help generate the NEC coat of appropriate

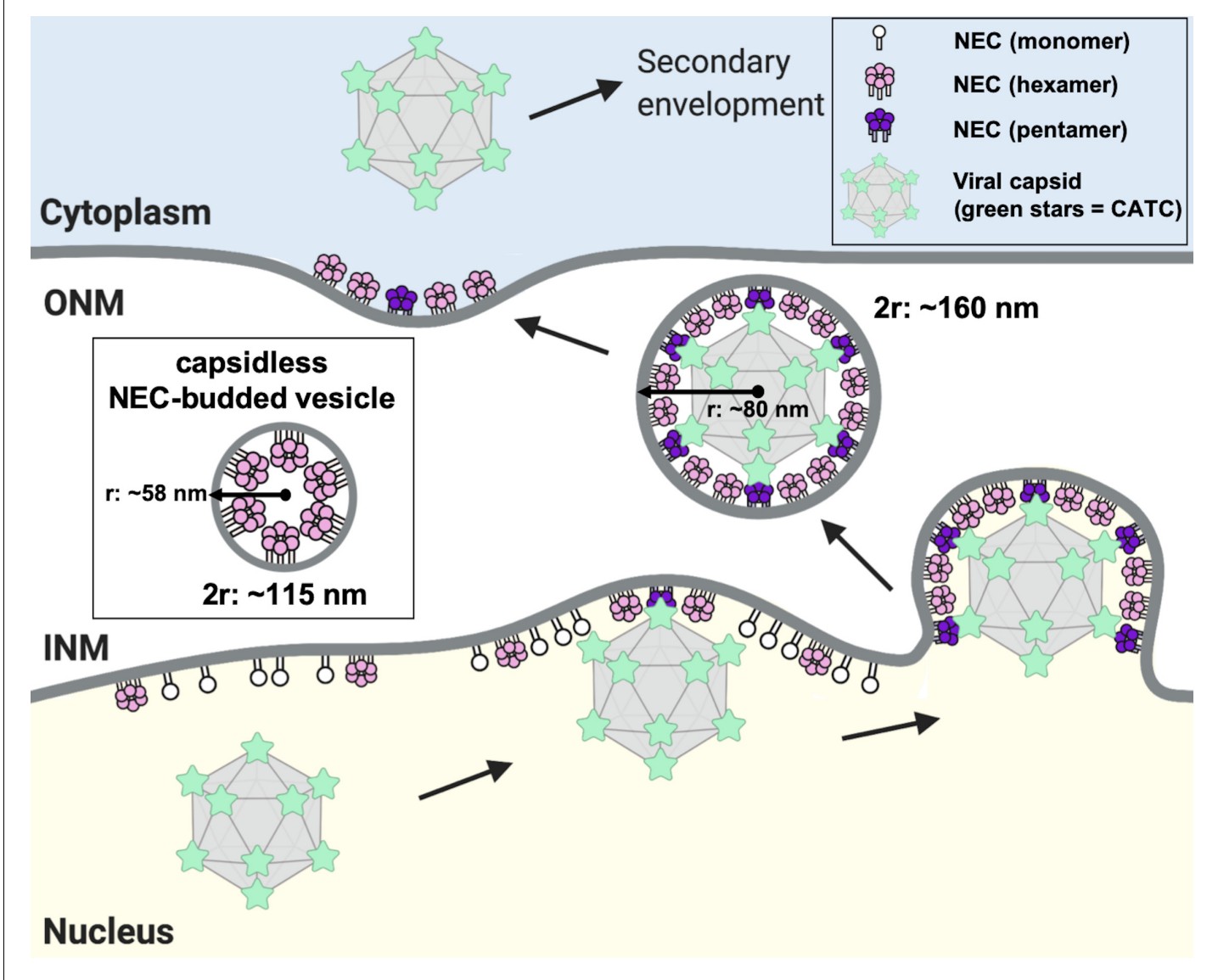

**Figure 8.** A model of NEC-mediated budding in HSV-1 infected cells. Capsid-bound UL25 induces the formation of pentagonal insertions (purple pentamers) within the NEC coat (pink hexamers and white monomers) as it is forming, which enables the formation of an NEC coat of appropriate size and curvature around the capsid. Inset shows a capsidless perinuclear vesicle formed in NEC-expressing uninfected cells that forms a hexagonal coat with presumably irregular defects, similar to the NEC coat formed in vitro. This figure was created with Biorender.com.

curvature as it assembles around the capsid. A similar strategy is observed during HIV-1 capsid formation by the Gag protein (*Mattei et al., 2016*). As the mature capsid is built, the Gag protein is cleaved, and the Gag capsid-domain builds a hexagonal lattice containing 12 pentamers to form a closed fullerene-like structure.

In vitro and in NEC-expressing cells, curved NEC coats are assembled even in the absence of a capsid, and we do not yet understand how the curvature can be achieved in these cases. Hexagonal NEC coats formed in in vitro or in NEC-expressing cells have a smaller diameter than those formed around the capsid (*Figure 8*), however, they may achieve coat curvature by other means, for example, by having irregular defects. Incorporation of irregular defects into curved hexagonal lattices has been observed for immature HIV capsids formed by Gag protein (*Briggs et al., 2009*; *Schur et al., 2015*) and in early poxvirus envelopes formed by the D13 protein (*Heuser, 2005*; *Hyun et al., 2011*). NEC could, potentially, use a similar strategy in the absence of a capsids.

## The NEC pentagon could nucleate the formation of the NEC coats

We further hypothesize that the first NEC pentagon formed upon initial contact with the capsid vertex could nucleate formation of the NEC coat (*Figure 8*), helping explain why in infected cells, NEC-mediated budding requires the capsid. Although in vitro or in NEC-expressing cells, the NEC budding does not require either the capsid or UL25, in both cases, the NEC is already in its active, uninhibited form. In contrast, during infection, the budding activity of the NEC is somehow inhibited until the capsid comes along. Binding to UL25 on the capsid could potentially serve as a trigger for releasing the NEC from its inhibited state, allowing budding to occur.

While UL25 is not required for NEC-mediated budding in vitro and in large amounts even inhibits it, small amounts of UL25 may be expected to stimulate budding in vitro by promoting formation of a limited number of NEC pentagons that could nucleate the coat assembly and promote coat curvature. Although we did not observe any stimulation of NEC-mediated budding by small amounts of UL25, this could be because in vitro, the NEC is already active. Perhaps, the stimulatory effect of UL25 only manifests itself with the inhibited NEC. In the future, the knowledge of the mechanism by which the virus inhibits the NEC budding activity during infection would allow one to test if UL25 alone could overcome this inhibition in vitro. Alternatively, this inhibition can only be overcome by the capsid.

## A model of NEC-mediated capsid budding during infection

Based on our observations, we propose the following model of NEC-mediated capsid budding during nuclear egress in infected cells (*Figure 8*). Binding of the cores of five neighboring copies of CATC at the capsid vertices to the NEC at the INM would promote formation of NEC pentagons, which would serve as a nucleation event for the assembly of the NEC coat around the capsid and also help anchor the capsid to the INM. As the hexagonal NEC coat continues to grow, the incorporation of pentagons into the coat at the points of contact with the vertices would both help attach the NEC coat to the capsid and introduce curvature into the NEC coat (*Figure 8*).

In both HSV-1 and PRV, removal of UL25 results in an accumulation of capsids at the INM, unable to undergo egress (*Klupp et al., 2006*; *Kuhn et al., 2008*). Our results suggest that UL25 both anchors the NEC coat to the capsid and contributes to formation of a curved coat. Additionally, our results could potentially explain why mostly mature, DNA-containing C-capsids undergo budding at the INM (*Klupp et al., 2011*; *Roizman and Furlong, 1974*). A- and B-capsids have fewer UL25 copies on the capsid surface (*Newcomb et al., 2006*), and we hypothesize that only C-capsids, which contain UL25 at a full occupancy, can generate pentagonal NEC insertions necessary for the formation of an NEC coat around the capsid. In this manner, NEC/UL25 interactions could provide a quality-control mechanism that would favor budding of mature, DNA-containing C-capsids – which have a full UL25 set – over the capsid forms with fewer UL25 copies thereby acting as a checkpoint during nuclear egress.

# Materials and methods

**Key resources table**

| Reagent type (species) or resource | Designation | Source or reference | Identifiers | Additional information |
|---|---|---|---|---|
| Gene (HSV-1 KOS) | UL25 | Geneart | JQ673480.1 | |
| Strain, strain background (*Escherichia coli*) | BL21(DE3) | Kerafast | LoBSTr | Chemically competent cells |
| Recombinant DNA reagent | eGFP-N2 (plasmid) | Clontech | eGFP-N2 | |
| Recombinant DNA reagent | pKH90 (plasmid) | PMID:24916797 | UL31 1–306 | |

*Continued on next page*

*Continued*

| Reagent type (species) or resource | Designation | Source or reference | Identifiers | Additional information |
|---|---|---|---|---|
| Recombinant DNA reagent | pJB02 (plasmid) | PMID:24916797 | UL34 1–220 | |
| Recombinant DNA reagent | pJB104 (plasmid) | This paper | UL25Δ44 | See Materials and methods, Cloning |
| Recombinant DNA reagent | pJB118 (plasmid) | This paper | NEC-CBM | See Materials and methods, Cloning |
| Recombinant DNA reagent | pJB123 (plasmid) | This paper | UL25Δ73 | See Materials and methods, Cloning |
| Recombinant DNA reagent | pED03 (plasmid) | This paper | UL25Δ44 Q72A | See Materials and methods, Cloning |
| Recombinant DNA reagent | pED05 (plasmid) | This paper | eGFP-UL25Δ73 | See Materials and methods, Cloning |
| Recombinant DNA reagent | pED13 (plasmid) | This paper | UL25Δ50 | See Materials and methods, Cloning |
| Recombinant DNA reagent | pED14 (plasmid) | This paper | eGFP-UL25Δ50 | See Materials and methods, Cloning |
| Peptide, recombinant protein | UL25Δ44 | This paper | | Purified from *E. coli* BL21(DE3) LoBSTr cells |
| Peptide, recombinant protein | UL25Δ44 Q72A | This paper | | Purified from *E. coli* BL21(DE3) LoBSTr cells |
| Peptide, recombinant protein | UL25Δ50 Q72A | This paper | | Purified from *E. coli* BL21(DE3) LoBSTr cells |
| Peptide, recombinant protein | UL25Δ58 Q72A | This paper | | Purified from *E. coli* BL21(DE3) LoBSTr cells |
| Peptide, recombinant protein | UL25Δ73 | This paper | | Purified from *E. coli* BL21(DE3) LoBSTr cells |
| Peptide, recombinant protein | eGFP-UL25Δ50 Q72A | This paper | | Purified from *E. coli* BL21(DE3) LoBSTr cells |
| Peptide, recombinant protein | eGFP-UL25Δ73 | This paper | | Purified from *E. coli* BL21(DE3) LoBSTr cells |
| Peptide, recombinant protein | NEC220 | This paper | | Purified from *E. coli* BL21(DE3) LoBSTr cells |
| Peptide, recombinant protein | NEC-CBM | This paper | | Purified from *E. coli* BL21(DE3) LoBSTr cells |
| Chemical compound, drug | Cascade Blue hydrazide | Thermo Fisher Scientific | Thermo Fisher Scientific: C687 | |
| Software, algorithm | ImageJ | ImageJ | RRID:SCR_003070 | |
| Software, algorithm | IMOD | IMOD | RRID:SCR_003297 | |
| Other | 1-palmitoyl-2-oleoyl-sn-glycero-3-phosphate | Avanti Polar Lipids | Avanti Polar Lipids:850857 | POPA |

*Continued on next page*

*Continued*

| Reagent type (species) or resource | Designation | Source or reference | Identifiers | Additional information |
|---|---|---|---|---|
| Other | 1-palmitoyl-2-oleoyl-glycero-3-phosphocholine | Avanti Polar Lipids | Avanti Polar Lipids:850457 | POPC |
| Other | 1-palmitoyl-2-oleoyl-sn-glycero-3-phospho-L-serine | Avanti Polar Lipids | Avanti Polar Lipids:840034 | POPS |

## Cloning

All primers used in cloning are listed in *Supplementary file 1*. Codon-optimized UL25 gene from HSV-1 strain KOS was synthesized by GeneArt. Digested PCR fragments encoding UL25Δ44 were subcloned by restriction digest into the pJP4 plasmid, which contains a His$_6$-SUMO-PreScission tag in frame with the BamHI restriction site of the multiple-cloning site in a pET24b vector, creating the pJB104 plasmid. DNA fragments encoding UL25Δ50 and UL25Δ73 were amplified by PCR from pJB104 (UL25Δ44) and subcloned into pJP4 by restriction digest using BamHI and XhoI, creating the UL25Δ50 (pED13) and UL25Δ73 (pJB123) plasmids. Site-directed mutagenesis of pJB104 yielded the UL25Δ44 Q72A mutant plasmid (pED03).

DNA encoding the eGFP sequence was PCR amplified out of the eGFP-N2 plasmid (Clontech) and subcloned via single-cut restriction digest into the corresponding UL25 plasmid harboring the cleavable His$_6$-SUMO tag [(either UL25Δ50 (pED13) or UL25Δ73 (pJB123)] creating either the eGFP-UL25Δ50 (pED14) or eGFP-UL25Δ73 (pED05) constructs.

Site-directed mutagenesis of pKH90 (UL31 1–306) using a splicing by overlap extension protocol *Heckman and Pease, 2007* followed by restriction digest into the pJP4 vector was used to create the UL31 D275A/C278A/K279A/D282A mutant (pJB118) used to produce the capsid-binding mutant of NEC220, NEC220-CBM.

## Expression and purification of NEC constructs

Plasmids encoding HSV-1 UL31 1–306 (pKH90) and UL34 1–220 (pJB02) were co-transformed into *Escherichia coli* BL21(DE3) LoBSTr cells (Kerafast) to generate NEC220 (*Bigalke et al., 2014*). Plasmids encoding HSV-1 UL31 1–306 D275A/C278A/K279A/D282A (pJB118) and UL34 1–220 (pJB02) were co-transformed into *E. coli* BL21(DE3) LoBSTr cells (Kerafast) to generate NEC220-CBM. All constructs were expressed using autoinduction at 37°C in Terrific Broth (TB) supplemented with 100 µg/mL kanamycin, 100 µg/mL ampicillin, 0.2% lactose and 2 mM MgSO$_4$ for 4 hr. The temperature was then reduced to 25°C for 16 hr. Cells were harvested at 5000 x g for 30 min. NEC220 proteins were purified as previously described (*Bigalke et al., 2014*) with slight modifications. The NEC220 and NEC220-CBM constructs were passed over 2 × 1 mL HiTrap Talon columns (GE Healthcare), rather than ion exchange as previously described, to remove excess cleaved His$_6$-SUMO before injection onto size-exclusion chromatography (as previously described).

## Expression and purification of UL25 constructs

Plasmids encoding either HSV-1 UL25 or eGFP-UL25 constructs were transformed into *E. coli* BL21 (DE3) LoBSTr cells and expressed using autoinduction at 37°C in TB supplemented with 100 µg/mL kanamycin, 0.2% lactose, and 2 mM MgSO$_4$ for 4 hr. The temperature was then reduced to 25°C for 16 hr. Cells were harvested at 5000 x g for 30 min. All purification steps were performed at 4°C. UL25 constructs were purified in lysis buffer (50 mM Na HEPES pH 7.5, 500 mM NaCl, 1 mM TCEP, and 10% glycerol). Cells were resuspended in lysis buffer supplemented with Complete protease inhibitor (Roche) and lysed with a microfluidizer (Microfluidics). The cell lysate was clarified by centrifugation at 13,000 x g for 35 min and was passed over Ni-NTA sepharose (GE Healthcare) column. The column was subsequently washed with 20 mM and 40 mM imidazole lysis buffer and bound proteins were eluted with 250 mM imidazole lysis buffer. The His$_6$-SUMO tag was cleaved for 16 hr

using PreScission Protease produced in-house from a GST-PreScission fusion protein expression plasmid. As a final purification step, UL25 constructs were purified with size-exclusion chromatography using either a Superdex 75 or 200 column (GE Healthcare) equilibrated with gel filtration buffer (20 mM Na HEPES, pH 7.0, 100 mM NaCl, and 1 mM TCEP). The UL25 constructs were purified to homogeneity as assessed by 12% SDS-PAGE and Coomassie staining. Fractions containing UL25 were concentrated up to ~30 mg/mL and stored at −80°C to prevent degradation observed at 4°C. Protein concentration was determined by absorbance measurements at 280 nm. The typical yield was 35 mg/L of TB culture.

## Co-sedimentation assay

Co-sedimentation of UL25Δ44 to acidic multilamellar vesicles (MLVs) was performed as previously described (*Bigalke et al., 2014*). MLVs were prepared in a 3:1:1 ratio of 1-palmitoyl-2-oleoyl-glycero-3-phosphocholine (POPC):1-palmitoyl-2-oleoyl-sn-glycero-3-phospho-L-serine (POPS):1-palmitoyl-2-oleoyl-sn-glycero-3-phosphate (POPA) (Avanti Polar Lipids). Background signal in the absence of liposomes is due to protein aggregation during centrifugation. The reported values represent the percentage (0–100%) of either NEC220 or UL25 bound to membranes. The standard error of the mean is reported for each measurement. Each condition was tested in two biological replicates each with two technical replicates. Biological replicates are experiments performed individually at separate times. Technical replicates are multiple repeats of the same experiment within a biological replicate.

## In vitro GUV budding assays

Giant unilamellar vesicles (GUVs; used for their large size and ease of identification at the microscope) were prepared as previously described (*Bigalke et al., 2014*). For NEC220 only budding quantification, a total of 10 μL of GUVs with a 3:1:1 ratio of POPC:POPS:POPA containing ATTO-594 DOPE (ATTO-TEC GmbH) at a concentration of 0.2 μg/μL was mixed with 1 μM NEC220 (final concentration), and 0.2 mg/mL (final concentration) Cascade Blue Hydrazide (ThermoFisher Scientific). For the NEC and UL25 titration experiments, 10 μL of GUVs and either 1, 6, 8, 10 or 20 μM of UL25Δ44 Q72A, UL25Δ58 Q72A or UL25Δ73 (final concentration) were incubated with 1 μM of NEC220 (final concentration) along with Cascade Blue. For NEC220-CBM and UL25 titration experiments, 10 μL of GUVs and either 1, 6, or 10 μM of UL25Δ44 Q72A (final concentration) were incubated with 1 μM of NEC220-CBM (final concentration) along with Cascade Blue. The total volume of each sample during imaging for all experiments was brought to 100 μL with gel filtration buffer and the reaction was incubated for 5 min at 20°C. Samples were imaged in a 96-well chambered coverglass. Images were acquired using a Nikon A1R Confocal Microscope with a 60x oil immersion lens at the Tufts Imaging Facility in the Center for Neuroscience Research at Tufts University School of Medicine. Images of NEC budding in the presence of eGFP-UL25 constructs were recorded after incubation of 10 μL of GUVs with 10 μM (final concentration) of either eGFP-UL25Δ50 Q72A or eGFP-UL25Δ73 and 1 μM of NEC220 (final concentration). Quantification was performed by counting vesicles in 15 different frames of the sample (~300 vesicles total). Raw data values for all experiments are given in each corresponding source data file. Each condition was tested in at least two biological replicates. Prior to analysis, the background was subtracted from the raw values. The reported values represent the average budding activity relative to NEC220 (100%). The standard error of the mean is reported for each measurement. Significance compared to NEC220 was calculated using an unpaired one-tailed *t*-test against NEC220.

## Isothermal titration calorimetry (ITC)

ITC measurements were recorded using a Microcal ITC200 (Malvern Panalytical) at the Center for Macromolecular Interactions at Harvard Medical School. A solution of UL25Δ44 (200 μM) was titrated into a solution of NEC220 (20 μM) in 20 mM Na HEPES, pH 7.0, 150 mM NaCl, 1 mM TCEP. Control experiments were performed by injecting UL25Δ44 into buffer. Thermograms were plotted by subtracting heats of the control experiments from the sample experiments. The data were not fit due to no detectable binding.

## Cryoelectron microscopy and tomography

A volume of 10 µL of a 1:1 mixture of 400 nm and 800 nm large unilamellar vesicles (LUVs) made of 3:1:1 POPC:POPS:POPA [prepared as previously described (*Bigalke et al., 2014*) were mixed on ice with a 30 µL solution of NEC220 and either UL25Δ44 Q72A or UL25Δ73, yielding an NEC:UL25 ratio of 1:10 (NEC220 concentration was at 1 mg/mL). After 30 min, 3 µL of sample was applied to glow-discharged (30 s) Quantifoil copper grids (R2/2, 200 mesh, Electron Microscopy Sciences), blotted on both sides for 4 s, and vitrified by rapid freezing in liquid ethane (Vitrobot). Grids were stored in liquid nitrogen until loaded into a Tecnai F20 transmission electron microscope (FEI) via a cryo holder (Gatan). The microscope was operated in low dose mode at 200 keV using SerialEM (*Mastronarde, 2005*) and images were recorded with a 4k × 4 k charge coupled device camera (Ultrascan, Gatan) at 29,000-fold magnification (pixel size: 0.632 nm). 2D cryo-EM images were recorded at defocus values of −4 to −8 µm and an electron dose ~15 e/Å$^2$. Images are displayed using ImageJ (RRID:SCR_003070) (*Schindelin et al., 2015*).

For single-axis cryoET data used to generate 3D EM data, samples were incubated on ice for 30 min, and 0.8 µL of 10 nm colloidal gold coated with protein A (Cell Microscopy Core, University Medical Center Utrecht, Department of Cell Biology) was added to the solution and mixed. The mixture (2.5 µL) was applied to freshly glow-discharged (30 s) Quantifoil R 3.5/1 grids (Electron Microscopy Sciences) and manually blotted before being flash-frozen in liquid ethane. Grids were loaded into an FEI Titan Krios electron microscope equipped with a Gatan imaging filter (GIF) and a Gatan K2 summit direct electron detection camera (Roper Technologies, Inc), operated at 300 kV. The acquisition for automated cryoET tilt series collection was performed using SerialEM (*Mastronarde, 2005*). A tilt series was collected in which the sample was tilted from 0° to +60° degrees and then from 0° to −60°, each in a stepwise fashion with 2° increments. Tilt series were acquired at a magnification of x53,000 (corresponding to a calibrated pixel size of 2.6 Å) with a maintained defocus value of −3 to −4 µm. The total electron dose was ~100 e/Å$^2$.

## 3D reconstruction and subtomographic averaging

The detailed steps of the 3D reconstruction and subtomographic averaging were previously described (*Imhof et al., 2019*; *Si et al., 2018*). Briefly, frames from each recorded tilt series were drift-corrected and averaged with *Motioncorr* (*Mastronarde, 2005*) and was further reconstructed with contrast transfer function (CTF) correction using the IMOD software package (RRID:SCR_003297) (*Kremer et al., 1996*). Two resulting tomograms were produced by the weighted back projection and simultaneous iterative reconstruction technique (SIRT) methods. High-contrast SIRT tomograms were binned 4x by the *binvol* program of *IMOD* to facilitate particle picking. A total of 1200 particles were picked manually in *IMOD* for tomograms containing LUVs, NEC220 and UL25Δ44 Q72A as follows. Distinct spike-like projections radiating outwards from the vesicle bilayer were assigned particles selected for analysis. Each spike was assigned two points (*head* and *tail*) in one contour where the *head* and the *tail* are the membrane-proximal and the membrane-distal ends of the density protruding from the vesicle. Only spikes of similar length were selected to ensure that each contained both NEC220 and UL25Δ44 Q72A. 3D sub-tomographic averaging was completed as described (*Imhof et al., 2019*; *Si et al., 2018*) using the PEET (particle estimation for electron tomography) software (*Nicastro et al., 2006*). Five-fold symmetry was only applied after five-fold symmetry was apparent in the averaged structure. The original dataset was split into two separate groups, even group and odd group, and averaged independently. Gold standard Fourier Shell Correlation (FSC) analysis for the averaged structure was performed by *calcUnbiasedFSC* in PEET when the two averaged structures converged. The reported resolution is 29 Å based on the 0.143 gold-standard FSC criterion. EM maps will be deposited into the Electron Microscopy Data Bank (EMDB) for immediate access upon publication.

## Acknowledgements

We thank Janna Bigalke for generating the UL25Δ44 and UL25Δ73 plasmids, purifying the corresponding proteins, performing the ITC and the size-exclusion experiments, and for generating the UL31-CBM mutant. We thank Alenka Lovy (Tufts University) for assistance with fluorescence microscopy experiments and Mike Rigney (Brandeis University) for assistance with cryoEM imaging. We

also thank Peter Cherepanov (Francis Crick Institute) for the gift of the GST-PreScission protease expression plasmid and Thomas Schwartz (Massachusetts Institute of Technology) for the gift of LoBSTr cells. ITC experiments were performed at the Center for Macromolecular Interactions in the Department of Biological Chemistry and Molecular Pharmacology at Harvard Medical School. Cry-oEM images were collected at the Electron Microscopy Facility at Brandeis University. CryoET data were collected at the Electron Imaging Center for Nanomachines at the University of California, Los Angeles. This work was funded by the NIH grants R01GM111795 (EEH), R01AI147625 (EEH), S10OD018111 (ZHZ), U24GM116792 (ZHZ), a Faculty Scholar grant 55108533 from Howard Hughes Medical Institute (EEH), NIH postdoctoral fellowship F32GM126760 (EBD), IRACDA postdoctoral fellowship K12GM133314 (EBD), the NSF grants DBI-1338135 (ZHZ) and DMR-1548924 (ZHZ), Burroughs Wellcome Fund Collaborative Research Travel Grant (EBD), the Natalie V Zucker Women Scholars Award (EBD), the UCLA Whitcome Pre-doctoral Fellowship (JZ), and the UCLA Dissertation Year Fellowship (JZ) .

## Additional information

### Funding

| Funder | Grant reference number | Author |
| --- | --- | --- |
| National Institutes of Health | R01GM111795 | Ekaterina E Heldwein |
| National Institutes of Health | S10OD018111 | Z Hong Zhou |
| National Institutes of Health | U24GM116792 | Z Hong Zhou |
| Howard Hughes Medical Institute | 55108533 | Ekaterina E Heldwein |
| National Institutes of Health | F32GM126760 | Elizabeth B Draganova |
| National Science Foundation | DBI-1338135 | Z Hong Zhou |
| National Science Foundation | DMR-1548924 | Z Hong Zhou |
| Burroughs Wellcome Fund | Collaborative Research Travel Grant | Elizabeth B Draganova |
| Natalie V. Zucker Women Scholars Award | | Elizabeth B Draganova |
| National Institutes of Health | R01AI147625 | Ekaterina E Heldwein |
| University of California, Los Angeles | Dissertation Year Fellowship | Jiayan Zhang |
| University of California, Los Angeles | Whitcome Pre-doctoral Fellowship | Jiayan Zhang |
| National Institutes of Health | K12GM133314 | Elizabeth B Draganova |

The funders had no role in study design, data collection and interpretation, or the decision to submit the work for publication.

### Author contributions

Elizabeth B Draganova, Conceptualization, Formal analysis, Funding acquisition, Validation, Investigation, Visualization, Methodology, Writing - original draft, Writing - review and editing; Jiayan Zhang, Formal analysis, Validation, Investigation, Visualization, Methodology, Writing - review and editing; Z Hong Zhou, Formal analysis, Supervision, Funding acquisition, Methodology, Project administration, Writing - review and editing; Ekaterina E Heldwein, Conceptualization, Supervision, Funding acquisition, Writing - original draft, Project administration, Writing - review and editing

### Author ORCIDs

Elizabeth B Draganova ⓘD https://orcid.org/0000-0003-3697-4774
Jiayan Zhang ⓘD https://orcid.org/0000-0003-3602-1199

Z Hong Zhou [ID] http://orcid.org/0000-0002-8373-4717
Ekaterina E Heldwein [ID] https://orcid.org/0000-0003-3113-6958

**Decision letter and Author response**
Decision letter https://doi.org/10.7554/eLife.56627.sa1
Author response https://doi.org/10.7554/eLife.56627.sa2

## Additional files

### Supplementary files
• Supplementary file 1. Supplementary Tables 1 and 2. UL25Δ44 Q72A/NEC particles used for cryo-oET averaging. List of primers used for cloning procedures described in Materials and methods.

• Transparent reporting form

### Data availability
The EM datasets have been deposited to the EMD under the reference numbers EMD-22207 and EMB-22208. All other data generated or analyzed during this study are included in the manuscript and supporting files. The source data for experiments presented in Figures 1, 2, and 3 are provided in Supplementary Table S2.

The following datasets were generated:

| Author(s) | Year | Dataset title | Dataset URL | Database and Identifier |
|---|---|---|---|---|
| Draganova EB, Zhang J, Zhou ZH, Heldwein EE | 2020 | EM maps | https://www.ebi.ac.uk/pdbe/entry/emdb/EMD-22207 | Electron Microscopy Data Bank, EMD-22207 |
| Draganova EB, Zhang J, Zhou ZH, Heldwein EE | 2020 | EM maps | https://www.ebi.ac.uk/pdbe/entry/emdb/EMD-22208 | Electron Microscopy Data Bank, EMB-22208 |

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
