## [Decision Letter]

**Acceptance summary:**

Herpesviruses assemble in the nucleus, but must then exit by budding through the inner nuclear membrane and then fusing with the outer nuclear membrane. Previous studies by this group and others have elegantly shown that these viruses encode nuclear egress complex (NEC) that can form hexameric cages that carry capsids through the inner nuclear membrane. However, those studies left open the question of how these cages can fully close because a hexagonal lattice alone cannot enclose space. Here, Draganova and colleagues present a satisfying solution to this mystery by showing that the HSV-1 capsid protein, UL25, promotes the formation of NEC pentagons (rather than hexagons), thereby allowing the NEC to form closed, curved cages that comprise a hexagonal lattice with 12 pentameric "defects" that form at the points of contact with the capsid vertices.

**Decision letter after peer review:**

Thank you for submitting your article "Structural basis for capsid recruitment and coat formation during HSV-1 nuclear egress" for consideration by *eLife*. Your article has been reviewed by two peer reviewers, and the evaluation has been overseen by a Reviewing Editor and Cynthia Wolberger as the Senior Editor. The reviewers have opted to remain anonymous.

The reviewers have discussed the reviews with one another and the Reviewing Editor has drafted this decision to help you prepare a revised submission.

Summary:

In this manuscript, Draganova and colleagues study how truncation variants of the HSV capsid protein UL25 binds to the nuclear egress complex in the presence of membranes. They conclude that UL25 and NEC interact on membranes – the two proteins form a net of pentagons on the membrane and this inhibits NEC mediated budding in vitro. Net formation and inhibition are dependent on the presence of residues 45-73. They speculate that UL25 on the capsid causes NEC to form pentamers to generate curvature as it assembles around the capsid.

Essential revisions:

This is a nicely performed study, with careful attention to detail and proper controls and analyses. Overall, the writing is clear and compelling, with the exception of part of the Discussion (see below). The data demonstrating an inhibitory effect of residues 45-73 on NEC mediated budding in vitro appears sound but there are some minor issues which should be addressed. The in vitro system is artificial, and a link from the in vitro system to the in vivo model is not demonstrated, but the system nevertheless appears to be revealing several key structural insights into the assemblies: i.e., regulation of NEC budding by UL25/capsid, and identification of pentameric assemblies that are likely vertices to help drive membrane curvature to form the coat. Therefore, has the potential to be an important step in the understanding this process at the molecular level.

The biggest concern is that although the "pentagon" model is attractive, it needs to be more strongly supported. In particular, the data showing the presence of pentagons is not yet fully convincing, as discussed below.

1) The authors show a structure of the pentagon star from NEC220+UL25d44 after applying five-fold symmetry. They say that five-fold symmetry is apparent before this is applied, but this should be convincingly shown and carefully tested because this is a main conclusion. Are they sure that apparent five-fold symmetry is not an artefact, and why? What fraction of the protein is forming pentagons – most of it or just some of it? Are there also hexagons, or only pentagons. Do they in fact have a mixture of states and how has this possibility been tested? It should be possible to sort this issue out by further analysis or image processing of existing data, without performing any new experiments.

2) As the authors point out, a network of stars would make a small spherical object. This would enclose a vesicle with a radius of about 10 nm. This is not what they see. What do they think is the overall arrangement? Some pentagons on the surface surrounded by five other pentagons and then something else in between? They say that they cannot map the coordinates of the sub-tomograms back onto the raw data but some analysis of the arrangement is needed to understand what is happening. There is an inconsistency between the proposed structure (a pentagon surrounded by five pentagons) and the vesicle morphology (not very curved).

---

## [Author Response]

Essential revisions:This is a nicely performed study, with careful attention to detail and proper controls and analyses. Overall, the writing is clear and compelling, with the exception of part of the Discussion (see below). The data demonstrating an inhibitory effect of residues 45-73 on NEC mediated budding in vitro appears sound but there are some minor issues which should be addressed. The in vitro system is artificial, and a link from the in vitro system to the in vivo model is not demonstrated, but the system nevertheless appears to be revealing several key structural insights into the assemblies: i.e., regulation of NEC budding by UL25/capsid, and identification of pentameric assemblies that are likely vertices to help drive membrane curvature to form the coat. Therefore, has the potential to be an important step in the understanding this process at the molecular level.The biggest concern is that although the "pentagon" model is attractive, it needs to be more strongly supported. In particular, the data showing the presence of pentagons is not yet fully convincing, as discussed below.1) The authors show a structure of the pentagon star from NEC220+UL25d44 after applying five-fold symmetry. They say that five-fold symmetry is apparent before this is applied, but this should be convincingly shown and carefully tested because this is a main conclusion. Are they sure that apparent five-fold symmetry is not an artefact, and why? What fraction of the protein is forming pentagons – most of it or just some of it? Are there also hexagons, or only pentagons. Do they in fact have a mixture of states and how has this possibility been tested? It should be possible to sort this issue out by further analysis or image processing of existing data, without performing any new experiments.

We now provide subtomographic averages prior to imposing symmetry that clearly show 5-fold symmetry within the NEC220 and UL25 layers (Figure 5—figure supplement 1). Granted that no systematic errors were introduced, we are confident that this is not an artifact. Indeed, by applying the same workflow to the coats containing only NEC220 (Figure 5B), we observed a different – 6-fold – symmetry, which essentially serves as a negative control. We have previously applied the same protocol to resolve the trimeric structure of glycoprotein gB from human cytomegalovirus both in its pre- and postfusion forms (Si et al., 2018) and an asymmetric structure of the 96-nm axonemal repeat of *Trypanosoma brucei* (Imhof et al., 2019).

Within the NEC220/UL25Δ44 dataset, we only observed 5-fold symmetry in the protruding densities, whereas in the NEC220 dataset, we observed only 6-fold symmetry (Figure 5B). As far as we can tell, when bound to UL25Δ44, the NEC220 forms only pentagons and not a mixture of pentagons and hexagons. We have included this description in the Results section of the manuscript (subsection “L25∆44 Q72A forms a net of stars bound to NEC pentagons”, first paragraph).

2) As the authors point out, a network of stars would make a small spherical object. This would enclose a vesicle with a radius of about 10 nm. This is not what they see. What do they think is the overall arrangement? Some pentagons on the surface surrounded by five other pentagons and then something else in between? They say that they cannot map the coordinates of the sub-tomograms back onto the raw data but some analysis of the arrangement is needed to understand what is happening. There is an inconsistency between the proposed structure (a pentagon surrounded by five pentagons) and the vesicle morphology (not very curved).

Visual analysis of the cryo-EM images (Figure 4—figure supplement 1) showed that UL25∆44/NEC220 did not completely cover the surface of the vesicles suggesting that it formed patches rather than full coats. Per reviewers’ suggestion, we mapped the coordinates of each sub-tomogram back onto the raw data and show that particles selected for analysis only partially cover the vesicles used for data processing (new Supplementary table 1 in Supplementary file 1). We have expanded the description of particle coverage in the last paragraph of the subsection “UL25∆44 Q72A forms a net of stars bound to NEC pentagons”.

While such partial coverage is in line with visual observations, we want to point out that the surface coverage by particles selected for analysis (new Supplementary table 1 in Supplementary file 1) is lower than the estimate based on visual inspection. This is because particles were selected for analysis according to a set of criteria to ensure the highest achievable quality of the reconstructions. Specifically, distinct spike-like projections radiating outwards from the vesicle bilayer were defined as particles selected for analysis. Each spike was assigned two points (*head* and *tail*) in one contour using IMOD where the *head* and the *tail* are the membrane-proximal and the membrane-distal ends of the density protruding from the vesicle. Only spikes of similar length were selected to ensure that each contained both NEC220 and UL25∆44 Q72A. We have added the description of the particle selection protocol to the Materials and methods section of the manuscript (subsection “3D reconstruction and subtomographic averaging”).

Regarding the overall arrangement, we hypothesize that the interactions between adjacent UL25∆44/NEC220 pentagons are not strong, which results in a loose network of pentagons forming patches instead of a small, spherical coats that would be formed if the arrangement were rigid. This interpretation is supported by the observation that the surrounding pentagons in the sub-tomographic averages (Figure 5) are not as sharply defined (lower resolution) as the central one (which contributed the most power to alignment during data processing), suggesting flexibility, likely due to weak interactions. We have added this explanation of the data to the manuscript (subsection “UL25∆44 Q72A forms a net of stars bound to NEC pentagons”, last paragraph and Discussion).